# Don't Throw Your Old Policies Away: Knowledge-based Policy Recycling Protects Against Adversarial Attacks

## Abstract

Recent work has shown that Deep Reinforcement Learning (DRL) is vulnerable to adversarial attacks, in which minor perturbations of input signals cause agents to behave inappropriately and unexpectedly. Humans, on the other hand, appear robust to these particular sorts of input variations. We posit that this part of robustness stems from accumulated knowledge about the world. In this work, we propose to leverage prior knowledge to defend against adversarial attacks in RL settings using a framework we call Knowledge-based Policy Recycling (KPR). Different from previous defense methods such as adversarial training and robust learning, KPR incorporates domain knowledge over a set of auxiliary tasks policies and learns relations among them from interactions with the environment via a Graph Neural Network (GNN). KPR can use any relevant policy as an auxiliary policy and, importantly, does not assume access or information regarding the adversarial attack. Empirically, KPR results in policies that are more robust to various adversarial attacks in Atari games and a simulated Robot Foodcourt environment.

## 1 Introduction

Despite significant performance breakthroughs in recent years (Mnih et al., 2015; Silver et al., 2016; Berner et al., 2019, e.g.,), Deep Reinforcement Learning (DRL) policies can be brittle. Specifically, recent works have shown that DRL policies are vulnerable to adversarial attacks — adversarially manipulated inputs (e.g., images) of small magnitude can cause RL agents to take incorrect actions (Ilahi et al., 2022; Chen et al., 2019; Behzadan & Munir, 2017; Oikarinen et al., 2021; Lee et al., 2021; Chan et al., 2020; Bai et al., 2018). To counter such attacks, recent work has proposed a range of defense strategies including adversarial training (Oikarinen et al., 2021; Behzadan & Munir, 2018; Han et al., 2018), robust learning (Mandlekar et al., 2017; Smirnova et al., 2019; Pan et al., 2019), defensive distillation (Rusu et al., 2016), and adversarial detection (Gallego et al., 2019a; Havens et al., 2018; Gallego et al., 2019a). While these defense methods can be effective, each has its limitations; adversarial training and adversarial detection require specific knowledge about the attacker. Robust learning adds noise during agent training, which can degrade performance (Tsipras et al., 2019; Yang et al., 2020). Defensive distillation is typically unable to protect against diverse adversarial attacks (Carlini & Wagner, 2016; Soll et al., 2019).

In this work, we explore an alternative defense strategy that exploits existing knowledge encoded in *auxiliary task policies* and known *relationships* between the policies. The key intuition underlying our approach is that existing task policies encode learnt low-level knowledge regarding the environment (e.g., possible observations, dynamics), whilst high-level specifications can provide guidance for transfer or generalization. Our approach is to leverage known and learnt relations between different policies as structural priors for an ensemble of policies; our hypothesis is that while a single task policy can be attacked, perturbing inputs such that multiple policies are negatively affected in a consistent manner is more difficult.

Our framework, which we call **Kno**wledge-based **P**olicy **F**usion (KPR), is partially inspired by the use of domain knowledge to address vulnerabilities to adversarial attacks in supervised learning (Melacci et al., 2021; Gürel et al., 2021; Zhang et al., 2022). In these works, domain knowledge is encoded as logical formulae over predicted labels and a set of features. A soft satisfiability score between

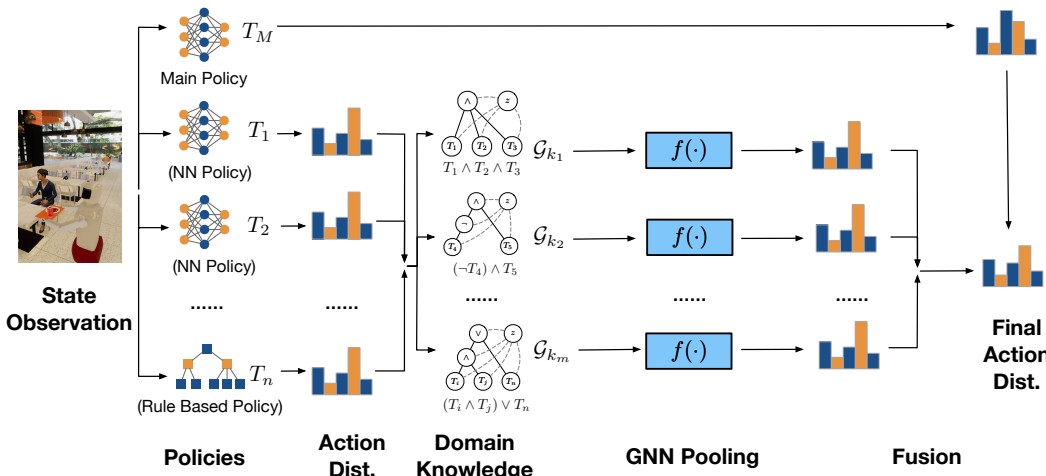

Figure 1: **K**nowledge-based **P**olicy **R**ecycling (KPR) Overview. Prior domain knowledge comprises a set of auxiliary tasks and a set of logical formulae defined over the auxiliary tasks. KPR combines these auxiliary tasks policies to obtain a new robust policy for the main task. At every time step, we obtain a state-conditioned action distribution for each auxiliary task, which are realizations of the variables in logical formulae. We represent logical formulae as graphs whose nodes are either auxiliary tasks or logical operators, e.g., $\neg, \wedge, \vee, \implies$. The instantiated logical formula graphs with node features are processed by a GNN pooling function (Section 3.2) to yield $m$ different action distributions (one for each graph). In other words, each logical graph results in a different policy. Finally, we combine/fuse these distributions (Section 3.3) to obtain a final action distribution. In summary, KPR uses the specified logical formula graph as prior structure and then learns flexible relations over auxiliary tasks from interaction data.

the predictions and given logic formulae is added to the objective to encourage the predictions to comply with the logical formulae. Even if part of the sample is corrupted by the adversary, the final output can be corrected by enforcing the domain knowledge rules. KPR extends this line of research to RL settings. Note that this extension is nontrivial; the auxiliary feature detectors used in supervised learning (Gürel et al., 2021; Zhang et al., 2022) do not capture temporal features, and more importantly, the consistency between the predictions and actions is not directly computable since the optimal actions for each state are unknown.

We address these issues by using auxiliary task policies and their relationships. In practice, these policies could be the intermediate by-products of curriculum/hierarchical learning or obtained via direct training with sub-goals. By combining these policies with specified and learnt relations, we construct an ensemble of policies on the target task. KPR uses graph neural networks (GNNs) as a backbone, which enables natural incorporation of graph-based domain knowledge while retaining the flexibility to learn from interaction data. The ensemble is then fused in a simple parameter-less manner to obtain a new robust task policy (Figure 1). From a practical perspective, KPR has a key advantage: it is both policy and attack agnostic. Specifically, KPR can utilize any type of policy representation (e.g., neural networks, rule-based policies) as either the main task policy or an auxiliary task policy. In addition, KPR doesn't require knowledge of the specific attack or access to the adversarial environment. Our empirical results show that KPR results in more robust policies across multiple attacks compared to baselines in a representative selection of Atari Games (Bellemare et al., 2013) and a high-dimensional Robot Food Court Environment (RoFoCo).

To summarize, this paper contributes **K**nowledge-based **P**olicy **R**ecycling (KPR), which leverages domain knowledge to defend against adversarial attacks in RL. Different from prior defense methods in reinforcement learning, such as adversarial training and robust learning, KPR is able to incorporate domain knowledge as structural prior and then learn flexible relations from interaction data. To the best of our knowledge, this is the first work to demonstrate that domain knowledge in the form of policies can be used to defend against adversarial attacks.

## 2 BACKGROUND AND RELATED WORKS

Similar to supervised learning, recent work has shown that DRL is also susceptible to adversarial attacks (Ilahi et al., 2022; Chen et al., 2019; Behzadan & Munir, 2017; Oikarinen et al., 2021).This area has drawn significant attention of late, and the following provides a brief overview; we refer readers desiring more detail to (Ilahi et al., 2022). Broadly speaking, there are two basic attack types depending on the assumed adversary: (i) **White-box attacks** (Goodfellow et al., 2015; Carlini & Wagner, 2017; Madry et al., 2018; Schwinn et al., 2021), where the adversary has perfect knowledge of the target model, which is the victim policy in the RL context, and (ii) **Black-box attacks** (Andriushchenko et al., 2020; Pomponi et al., 2022), where the adversary does not know the model nor any of its attributes. Similar to previous works (Huang et al., 2017; Behzadan & Munir, 2017; Pattanaik et al., 2018; Zhang et al., 2020; Sun et al., 2022), we assume that the attacker does not have the ability to change the environment directly but perturbs the state observations returned by the environment before the agent observes them.

Existing defense methods can be categorized into: (i) **Adversarial training** (Oikarinen et al., 2021; Behzadan & Munir, 2018; Han et al., 2018), where the RL agent is exposed to the adversarial environment during training (ii) **Robust learning** (Mandlekar et al., 2017; Smirnova et al., 2019; Pan et al., 2019), which is a training mechanism to ensure robustness against training-time adversarial attacks. A common approach is to add noise to the parameter state while training; (iii) **Adversarial detection** (Gallego et al., 2019a; Havens et al., 2018; Gallego et al., 2019a) trains a separate model to detect adversarial input and contaminated input (e.g., image frames) are replaced with predicted/generated versions; (iv) **Policy distillation** (Czarnecki et al., 2019) focuses on transferring knowledge from one or multiple policies to the target policy in a student-teacher framework.

KPR is different from the above strategies. Unlike adversarial training and variants of adversarial detection (Gallego et al., 2019b; Lin et al., 2017), we do not assume knowledge of the attack. In contrast to robust learning, KPR does inject noise during training. Although both policy distillation and KPR can fuse multiple policies, the methodology and application are different. Policy distillation does not leverage the relation between input policies and thus does not explicitly encourage structural consistency, which limits its defense performance as found in (Carlini & Wagner, 2016; Soll et al., 2019) and in our experiments (Section 4). KPR can be seen as a policy ensemble (Wiering & van Hasselt, 2008), albeit one that leverages prior knowledge in its construction. Prior works have suggested that ensemble methods offer some protection against adversarial attacks in supervised settings (Wiering & van Hasselt, 2008). However, to our knowledge, policy ensembles have yet to be used as a defense strategy in RL.

## 3 KNOWLEDGE-BASED POLICY RECYCLING (KPR)

As in standard RL, we consider a discounted discrete-time Markov Decision Process (MDP) $(\mathcal{S}, \mathcal{A}, R, \mathcal{T}, \gamma, d_0)$, where $\mathcal{S}$ is a set of states, $\mathcal{A}$ is a set of discrete action, $R : \mathcal{S} \times \mathcal{A} \to \mathbb{R}$ is the reward function, and $\mathcal{T} : \mathcal{S} \times \mathcal{A} \to \mathcal{P}(s)$ is the transition function, $\gamma \in [0, 1]$ is the discount factor, and $d_0 \in \mathcal{P}(s)$ is the distribution over the initial state. At time step $t$, the agent is in the state $s_t \sim \mathcal{T}(s_t, a_t)$ and receives a reward $R(s_t, a_t)$. The agent's objective is to learn a policy $\pi : \mathcal{S} \to \mathcal{P}(A)$ that maximizes the expected cumulative rewards,

$$\max_{\theta} J(\pi_\theta) = \mathbb{E}_{s_0 \sim d_0, s_{t+1} \sim \mathcal{T}(s_t, \pi_\theta(s_t))} \left[ \sum_{t=0}^{\infty} \gamma^t R(s_t, a_t) \right] \tag{1}$$

We consider a setting where the policy is trained in a benign environment and then deployed to a test environment that may be adversarial.

Our goal is to retain expected cumulative rewards in the presence of a white box attacker that aims to alter our agent's actions by perturbing observations $s'_t = s_t + \delta_t$, s.t. $\delta \in \Delta$, where $\Delta$ is a perturbation set, e.g., an $\ell_2$ ball around $s$ with $\epsilon$ radius, i.e. $\ell_2(\delta_t) \leq \epsilon, \forall t \geq 0$. We consider a setting where we do *not* have access to the attacker's perturbation set nor how it is optimizing. In this information impoverished setting, our defensive options are relatively limited. One approach is to sample possible attackers (with various perturbation abilities) and train a policy to be robust against these attacks. However, this approach is computationally expensive.

We posit the susceptibility of a policy to an attacker is due principally to overfitting on a specific task. To alleviate this issue, we propose to leverage prior knowledge — comprising auxiliary tasks and relations between them — that can enable better robustness to noisy or perturbed observations. Let us define our main task as $T_M$ (which is modeled by an MDP as above) and an associated main task policy $\pi_{T_M}$ that maximizes returns. We assume we possess a main task policy trained in the benign environment. We also have access to $n$ auxiliary task policies $\Pi_{aux} = \{\pi_{T_1}, \pi_{T_2}, \ldots, \pi_{T_n}\}$ for tasks $\Upsilon_{aux} = \{T_1, T_2, \ldots, T_n\}$. Each task is modeled by an MDP tuple with the same elements as $T_M$ but with a different reward function. We define the *relation set* $\mathcal{I} = \{\mathcal{K}, f(\cdot)\}$ comprising *explicitly* specified relations between the task policies $\mathcal{K}$ (e.g., from a domain expert) and learnt knowledge that will be represented *implicitly* by the function $f(\cdot)$.

While prior knowledge usually enters into training as a regularization term in the loss, we incorporate this knowledge *directly* into the policy. We seek to obtain an augmented policy $\widehat{\pi}_{T_M}$ that maximizes returns and is conditioned upon $\Pi_{aux}$ and $\mathcal{I}$. The challenge is to ensure that this augmented policy has sufficient capacity to perform well on the task yet be robust to attacks. Our approach is an ensemble method where we use $\mathcal{I}$ and $\Pi_{aux}$ to construct a new set of policies $\widehat{\Pi}_{T_M}$ for the main task $T_M$ (Fig. 1). We then combine these policies using a simple parameter-less fusion mechanism,

$$\widehat{\pi}_{T_M}(s) = p(a|s, \Pi_{aux}, \mathcal{I}) = p(a|\widehat{\Pi}_{T_M}, s). \tag{2}$$

In the following subsections, we provide details on the above: we will first discuss how prior domain knowledge regarding existing tasks can be specified. Next, we detail a Graph Neural Network (GNN) pooling network to process knowledge and state information to learn relations. Finally, we discuss the simple voting mechanism to derive the final policy action distribution.

### 3.1 SPECIFYING RELATIONS VIA LOGICAL GRAPHS

In our setting, domain knowledge comprises (i) the set of auxiliary tasks policies given, $\Pi_{aux}$, which could be any type of policy (including non-differentiable or rule-based policies), and (ii) the logical relations among them. We focus on propositional logic, which is well-defined and unambiguous compared to natural language, yet relatively easy for humans to derive and interpret. A proposition $p$ is a statement which is either True or False. A formula F is a compound of propositions connected by logical connectives, e.g., $\neg, \wedge \vee, \implies$. A logical formula (and corresponding truth assignments) can be represented as a undirected graph $\mathcal{G} = (\mathcal{V}, \mathcal{E})$ with nodes $v_i \in \mathcal{V}$, and edges $(v_i, v_j) \in \mathcal{E}$. Individual nodes are either propositions (leaf nodes) or logical operators $(\neg, \wedge, \vee)$, where subjects and objects are connected to their respective operators.

In our work, leaf nodes are auxiliary tasks, which are True if the task is successful and False otherwise, where *success* of the task could be defined as whether the accumulated return attains a certain threshold. For example, the logical relation formula $T_M \implies T_1 \wedge T_2$ expresses that "if the the main task is successful, then both the auxiliary tasks $T_1$ and $T_2$ are successful". The features of the leaf nodes are the predicted action distributions conditioned on the state $s_t$; the features of the logical operators are fixed to randomly generated vectors. To incorporate state information, we add a state feature node to every logical relation graph and connect it to every other node. As state features, we use the latent vector $z_s$ obtained from a simple VAE (Kingma & Welling, 2014) trained to reconstruct the state $s$.

The set of logical relations forms the set $\mathcal{K} = \{k_i(\Upsilon_{aux})\}_{i=1}^{m}$ defined over the auxiliary tasks $\Upsilon_{aux}$ where each $k_i$ is represented as a graph $\mathcal{G}_{k_i}$. Note that the logical relations need not to be complete nor error-free; KPR can tolerate a degree of misspecification as relations are also learnt from interaction data.

### 3.2 LEARNING RELATIONS FROM INTERACTION DATA

**Graph Neural Network Pooling Function.** To utilize the structural information contained in the logical relation graphs and enable learning from interaction data, we adopt Graph Neural Networks (GNNs) (Fey & Lenssen, 2019). GNNs pass and aggregate the messages from the neighbors to encode the nodes in the graph with learnable weights. To obtain a single action distribution from each logical relation graph, we add a graph pooling layer that maps the entire graph into a compact representation. We adopt Graph Multiset Pooling (Baek et al., 2021), which satisfies permutation

invariance, as our GNN pooling function $f(\cdot)$. The accumulated knowledge $\mathcal{I}$ consists of both the logical relations graphs and learnt weights of the GNN pooling function $f(\cdot)$.

Graph Multiset Pooling network $f(\cdot)$, consists of a two-layer message-passing module $g(\cdot)$, a graph multi-head attention pooling module that condenses all nodes to representative nodes $h_l(\cdot)$, a multi-head self-attention module $q_s(\cdot)$, and the second multi-head attention pooling module that condenses the entire graph to one vector $h_1(\cdot)$. Put together,

$$f(\mathcal{G}_{k_i}) = h_1\Big(q_s\Big(h_l\big(g(\mathcal{G}_{k_i})\big)\Big)\Big),\tag{3}$$

where $\mathcal{G}_{k_i}$ is the input logical graph constructed from the logical relation formula $k_i$.

The message-passing function updates node representations by aggregating its neighbors' messages. In particular, the message-passing function we use is:

$$x_v^{(l)} = \sum_{u\in\mathcal{N}\cup\{v\}} \frac{1}{\sqrt{D(v)}\sqrt{D(u)}}\big(W^T x_u^{(l-1)}\big),\tag{4}$$

where $x_v^l$ is the node representation of node $v$ at $l^{\text{th}}$ step. $\mathcal{N}(v)$ denotes a set of neighboring nodes of $v$, $W$ is the weight matrix, and $D(v), D(u)$ denotes the degree of node $v, u$ respectively.

An attention function $q$ can be described as mapping a query and a set of key-value pairs to an output, where the query $Q$, keys $K$, values $V$, and output are all vectors. The output is computed as a weighted sum of the values, where the weight assigned to each value is computed by a compatibility function of the query with the corresponding key.

$$q(Q, K, V) = \text{Softmax}\left(\frac{QK^T}{\sqrt{d_l}}\right)V.\tag{5}$$

The multi-head attention function $q^{\text{multi}}$ simply uses a multiple attention function with the linearly projected query, keys, and values matrices. To explicitly use the graph structure in multi-head attention, the keys and values are generated using one layer of message-passing function instead of conventional linear projection.

$$h_k\big(g(\mathcal{G}_{k_i})\big) = I\Big(q^{\text{multi}}\Big(Z, g^K\big(g(\mathcal{G}_{k_i})\big), g^V\big(g(\mathcal{G}_{k_i})\big)\Big)\Big),\tag{6}$$

where $I(\cdot)$ denotes the residual connection function, $q^{\text{multi}}(\cdot)$ is the multi-head attention function, $Z$ is a parameterized seed matrix that is optimized end-to-end acting as the query matrix, and $g^K, g^V$ represents the message-passing functions for key and value respectively.

**Model Training.** Each $f(\mathcal{G}_{k_i})$ can be interpreted as a policy in an ensemble; $f$ takes a logical formula graph as input, and outputs an action distribution, i.e., $p_{k_i}(a|s) = f(\mathcal{G}_{k_i})$. The node feature for the auxiliary task node is the respective predicted action distribution and the auxiliary task identifier. The node features for logical operators are fixed randomly generated vectors that share the same dimension with auxiliary task features. To learn the relationships between tasks, we train each ensemble policy to predict the action with the highest probability under the main task policy. More precisely, we train the graph neural network pooling function $f$ with the following loss:

$$L_f = \mathbb{E}_{\tau\sim p(\tau|\pi_{\text{T}_M})}\left[\sum_{t=0}^{T}\sum_{i=0}^{m} \text{CrossEntropy}\Big(\pi_{\text{T}_M}(s_t), f\big(k_i, \Pi_{aux}(s_t), z_{s_t}\big)\Big)\right],\tag{7}$$

where $\tau = \{(s_t, a_t, R(s_t, a_t)\}$ is a trajectory sampled using the main task policy $\pi_{\text{T}_M}$ and $z_{s_t}$ are the state features. Note that KPR does not need to access the perturbed environment during training.

### 3.3 Policy via Fusion of Action Distributions

At this stage, we have $m$ action distributions $p_{k_i}(a|s)$ for each logical relation $k_i$. To obtain a final action distribution, we will need to combine or fuse these action distributions together. We adopt a very simple voting-like mechanism that requires *no training*.

The fusion is performed via two main steps: (i) task policy filtering and (ii) action counting. We first select the top-3 action distributions by selecting the most "confident" models as measured by negative entropy. Intuitively, this filters away irrelevant action distributions given the current state.

Next, we form a new action distribution by vote counting. Each of the three remaining policies casts a positive vote for their top scoring action (with largest $p(a_j|s)$) and a negative vote for their lowest scoring action (smallest $p(a_j|s)$). For each action $a_j$, we tally the number of positive votes $o_j^+$ and negative votes $o_j^-$, and construct a new action distribution for the main task $T_M$:

$$\widehat{\pi}_M = p(a_j) = \frac{o_j^+ - o_j^-}{|\mathcal{A}|} \tag{8}$$

## 4 EXPERIMENTS

In this section, we focus on validating that incorporating knowledge offers protection against adversarial attacks in deep RL. Specifically, we conduct experiments on a selection of Atari games and a high-dimensional Robot Food Court Environment (RoFoCo)[1].

**Compared Methods.** We compared our method with policy ensemble (Wiering & van Hasselt, 2008), policy distillation (Rusu et al., 2016), and adversarial training (Goodfellow et al., 2015) as strong baselines[2]. For Policy Ensemble, we use five main task policies (trained with different random seeds) with majority voting. Since we focus on attack-agnostic defense methods, we conduct adversarial training over a union of commonly-known attacks, including the attacks that we evaluate on; this method approximates an "attack agnostic" adversarial training. Recall that the domain knowledge $\mathcal{I}$ comprises the relations among the auxiliary tasks and the main task. To investigate the role $\mathcal{I}$ plays, we included a variant of our method, MLP Fusion, which replaces the GNN with a MLP whose input are the action distributions of the auxiliary policies.

**Attacks.** In Atari games, we used four common white-box attacks, FGSM (Goodfellow et al., 2015; Huang et al., 2017), PGD (Madry et al., 2018) and Jitter (Schwinn et al., 2021); and a black-box attack, Square (Andriushchenko et al., 2020). In the Robot Food Court environment (RoFoCo), we selected FGSM and PGD as white-box attacks (since they caused the most significant performance degradation in the Atari games experiments) and Square as the black-box attack. The attacks were implemented using the `torchattacks` package (Kim, 2020).

### 4.1 ATARI GAMES

**Environment.** We evaluated KPR on three Atari games: Road Runner, River Raid, and Space Invader, which are representative tasks that can be naturally decomposed into auxiliary tasks, e.g., collect targets, shoot enemies, and avoid collisions. For each game, the state is a stack of four consecutive frames, where each frame is pre-processed to size $84 \times 84$. Detailed environment settings can be found in Appendix B.2.

**Main Task Policy.** We adopt PPO (Schulman et al., 2017; Huang et al., 2022) as the learning algorithm. Each policy is trained with 10 million frames. Alternative algorithms and policies such as DQN (Mnih et al., 2015) and Rainbow DQN (Hessel et al., 2018) can be used without changing the overall proposed framework.

**Auxiliary Tasks and Domain Knowledge.** Due to space restrictions, complete auxiliary tasks information and domain knowledge relations are in Appendix B.2. Taking Road Runner as an example, the auxiliary tasks are: "$T_1$: Collect the bird seeds on the road." and "$T_2$: Avoid the cars.". Denote the main task, "Collect the bird seeds while avoiding colliding with cars on the road", as $\pi_{T_M}$. The logical relation is $T_M \implies T_1 \wedge T_2$.

---

[1]We also illustrate KPR using a goal-finding grid-world environment as a proof-of-concept. Due to space constraints, please refer to Appendix A for more details.

[2]We also attempted to compare other methods, i.e., state-adversary DQN (SA-DQN) (Zhang et al., 2020; 2021), Policy Adversarial Actor Director (PA-AD) (Sun et al., 2022), Adversary Agnostic Policy Distillation (A2PD) (Qu et al., 2021). However, these methods do not perform sufficiently well in our environments, and thus, we exclude them from our comparison.

Table 1: Test-time episode accumulated returns under three Atari games. The returns are averaged over 70 episodes and standard errors are reported in brackets. The best scores are in bold.

| Algorithm | Game | Benign | Attack Methods | | | |
| | | | FGSM | PGD | Jitter | Square |
|---|---|---|---|---|---|---|
| Main Task Policy | Road Runner | 41388.57 (2080.28) | 16764.29 (854.49) | 15351.43 (918.47) | 16904.29 (672.62) | 40467.14 (1735.77) |
| Ensemble | | 42548.57 (1802.11) | 31437.14 (1724.59) | 30970.00 (951.13) | 22495.71 (1173.65) | 37414.29 (1864.43) |
| Policy Distillation | | 40072.86 (1701.94) | 14931.43 (803.47) | 15891.43 (888.98) | 17501.43 (1052.50) | 40467.14 (1735.77) |
| Adversarial Training | | 25762.86 (1250.54) | 6032.86 (350.16) | 5538.57 (326.89) | 8252.86 (433.16) | 25922.86 (1247.81) |
| MLP Fusion | | 37311.43 (1449.47) | 21752.86 (1292.49) | 21502.86 (1101.42) | 18524.29 (1161.33) | 41310.00 (1191.39) |
| KPR (Ours) | | **43498.57** (2203.07) | **44417.14** (3526.12) | **34427.14** (2301.96) | **34187.14** (1673.32) | **46128.57** (1707.11) |
| Main Task Policy | River Raid | 7491.57 (149.04) | 2318.00 (75.87) | 3567.29 (190.00) | 5589.43 (231.55) | 7005.00 (131.70) |
| Ensemble | | 7907.14 (113.39) | 2178.43 (61.28) | 2752.86 (97.65) | 5107.86 (181.46) | 4700.43 (294.33) |
| Policy Distillation | | 7948.71 (104.53) | 2438.71 (91.36) | 3575.57 (169.24) | 5719.57 (211.32) | 5943.14 (291.97) |
| Adversarial Training | | 5951.14 (137.67) | 2631.14 (117.13) | 3230.71 (134.76) | 5172.29 (159.03) | 2846.43 (155.60) |
| MLP Fusion | | 7832.29 (57.62) | 1927.00 (57.55) | 2295.29 (53.45) | 5154.29 (218.38) | 7626.00 (65.32) |
| KPR (Ours) | | **7966.14** (65.24) | **2720.14** (113.61) | **3683.57** (186.70) | **6470.29** (186.24) | **7807.14** (83.37) |
| Main Task Policy | Space Invaders | 1022.57 (55.97) | 537.93 (21.74) | 547.07 (20.62) | 832.00 (45.82) | 976.50 (59.22) |
| Ensemble | | **1166.64** (53.47) | 621.79 (25.80) | 649.14 (26.75) | 883.71 (42.71) | 661.21 (56.41) |
| Policy Distillation | | 1069.14 (53.30) | 571.93 (25.77) | 601.71 (29.60) | 759.57 (43.00) | 711.14 (60.52) |
| Adversarial Training | | 618.64 (27.66) | 454.71 (29.16) | 357.64 (23.46) | 591.86 (28.01) | 356.29 (20.03) |
| MLP Fusion | | 1059.07 (53.60) | 733.00 (32.23) | 696.71 (34.08) | 875.07 (44.94) | 765.00 (24.64) |
| KPR (Ours) | | 1161.64 (61.59) | **810.36** (43.82) | **817.21** (45.74) | **938.64** (48.70) | **1009.71** (55.51) |

**Results and Discussion.** Our experimental results are summarized in Table 1, which shows test-time episode returns averaged over 70 episodes with standard errors in brackets. In each game, the main policy solved the task but performance degraded when attacked, as expected. In general, KPR improves robustness to all five attacks across the three environments. There is some performance degradation but it is less severe compared to the unprotected main task policy. This is most apparent for the strongest attacks, i.e., FGSM and PGD.

Interestingly, we see that adversarial training led to poorer policies in the benign environment and was also generally ineffective against FGSM and PGD. We believe this is due to the attack-agnostic training scheme; in typical use, adversarial training requires knowledge of the attacker but in our experiments, different attackers were sampled and this led to significant noise that hampered training.

Policy Ensemble, MLP Fusion, and KPR are all ensemble methods, but differ in their construction. Although not specifically designed for defense, the simple policy ensemble is surprisingly effective in the Road Runner domain. Policy Ensemble and MLP fusion achieve comparable performance, but are poorer compared to KPR. These results suggest that using existing prior knowledge in the form of policies and logical relations do result in more robust policies.

**Robot Food Court Environment (RoFoCo)**

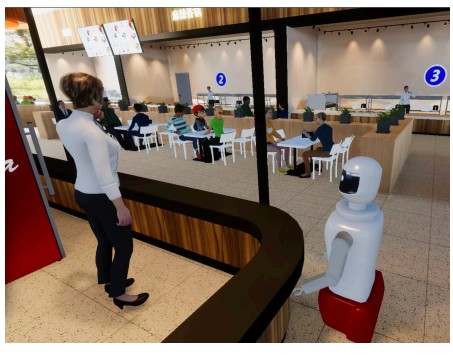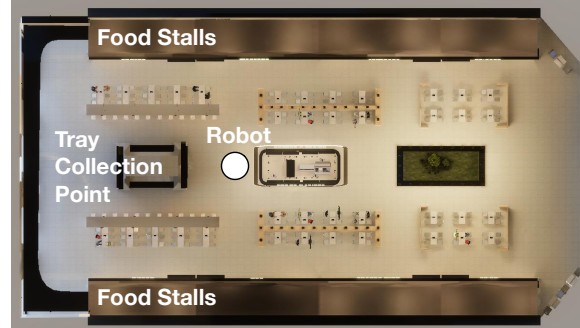

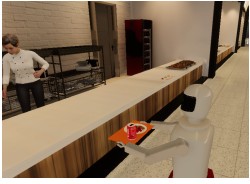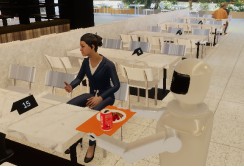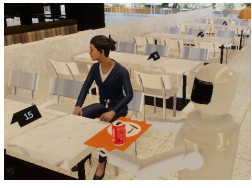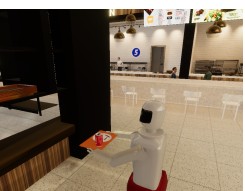

(1) Collect food from the correct food stall

(2) Deliver food to the correct table

(3) Collect the used tray after customers finished eating

(4) Deliver the used tray to the tray collection point

Figure 2: Robot Food Court Environment (RoFoCo) and a breakdown of the main task into four steps.

## 4.2 ROBOT FOOD COURT ENVIRONMENT (ROFOCO)

**Environment.** This environment simulates a service robot in a food-court setting and was developed using Unity (Juliani et al., 2018). The agent is a food serving robot. A food stall number and a table number are given at the beginning of every episode. The main task is to collect food from the instructed food stall and deliver it to the instructed table number. After the customer finishes eating, the robot should pick up the used tray and deposit it at the tray collection point. The task is divided into four stages as illustrated in Figure 2. The agent receives a $+10$ reward for completing each intermediate stage and an additional $+20$ reward for completing the whole task. The agent will get a $-0.1$ penalty at every time step and an additional $-5$ penalty if it tries to incorrectly perform pick up or put down actions on objects, such as trying to pick up food while the customer is eating. The maximum number of steps is 1,000. The available actions are: 1) Move forward; 2) Move backward; 3) Turn left 90 degrees; 4) Turn Right 90 degrees; 5) Pick up; 6) Put down; 7) Do nothing. The observation at each time step is a $128 \times 128$ RGB image.

**Main Task Policy.** As the observation space is significantly larger than the Atari games, exploring from scratch is computational resource consuming. To obtain the main task policy, we first initialize the agent with the policy obtained using imitation learning (Hussein et al., 2017) on a set of expert demonstrations. DQN (Mnih et al., 2015) is used to refine the policy via interactions further.

**Auxiliary Tasks and Domain Knowledge.** Due to space constraints, we give a few examples of auxiliary tasks and domain knowledge relations here and provide the complete list in Appendix B.3. Three example auxiliary tasks are 1) $T_1$: Navigate to the food stall; 2) $T_2$: Pick up food from customer tables; and 3) $T_3$: Pick up food from the food stall. Denote the main task we described as $T_M$. One example relation is $T_M \implies T_1 \wedge (\neg T_2) \wedge T_3$.

**Results and Discussion.** Test-time performance comparison is summarized in Table 2, which shows episode returns averaged over 50 test episodes. KPR outperforms other baselines across attacks, which further supports the notion that prior policy knowledge encourages robustness in policies. As before, KPR policies experienced less severe degradation against the various attacks (which are unknown at training-time). As in the Atari games, we observed that the ensemble methods are able to provide comparably effective protection compared to policy distillation and adversarial training.

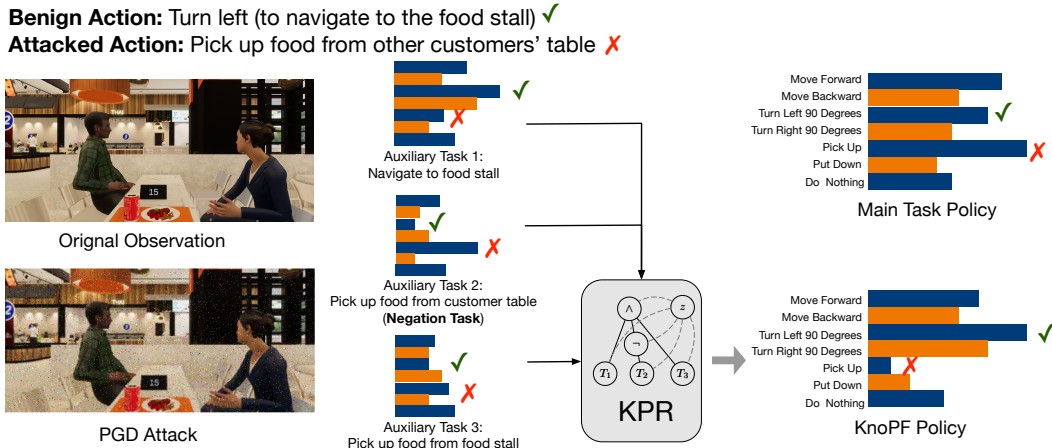

Figure 3: Adversarial Attacks Illustration. The adversary successfully misled the agent into picking up food from other customers' tables. KPR corrects the final action distribution prediction by leveraging auxiliary policies and domain knowledge.

Table 2: Test-time episode accumulated returns under the Robot Food Court environment. The returns are averaged over 50 episodes and standard errors are reported in brackets. The best scores are in bold.

| Algorithm | Benign | Attack Methods | | |
| --- | --- | --- | --- | --- |
| | | FGSM | PGD | Square |
| Main Task Policy | 39.72 (2.62) | 28.26 (3.46) | 11.22 (5.21) | 27.04 (3.66) |
| Ensemble | 40.50 (2.08) | 30.26 (2.67) | 10.18 (5.35) | 26.92 (3.42) |
| Policy Distillation | 42.35 (1.78) | 19.33 (5.27) | 15.23 (4.35) | 19.10 (3.85) |
| Adversarial Training | 36.28 (2.81) | 18.98 (6.16) | 6.14 (4.49) | 22.22 (3.24) |
| MLP Fusion | **43.40** (1.73) | 24.57 (3.85) | 19.00 (3.87) | 26.89 (3.95) |
| KPR (Ours) | 42.13 (2.17) | **34.70** (3.07) | **28.51** (4.14) | **31.62** (4.42) |

As a qualitative comparison, Figure 3 shows an example where the adversary successfully misled the robot into picking up food from other customers' tables. By leveraging auxiliary policies, such as navigating to the food stall and picking the food from the food stall, together with the relations between the tasks, KPR is able to correct the final action distribution prediction.

## 5 CONCLUSION AND FUTURE WORK

This paper proposes KPR, a novel approach to leverage domain knowledge to defend against adversarial attacks in reinforcement learning settings. KPR incorporates domain knowledge from auxiliary policies and specified logical relations between tasks, then learns flexible relations from interaction data via graph neural networks. The main advantage of KPR is that it is both policy and attack agnostic; any type of policy could be utilized, and no access nor information about the attack is required. We demonstrated its efficacy empirically in both Atari games and the complex Robot Food Court environment (RoFoCo).

A number of promising avenues exist for future research. In this work, we mainly experimented with neural network policies and future work can look into other auxiliary and main policies (e.g., interpretable rule-based policies). Next, KPR worked well in RoFoCo, which is a highly-complex environment, but it is necessary to test KPR (and other existing defense methods) in real-world environments. Finally, we believe that KPR can also defend against alternative threat models, including observed adversaries that comply with environmental constraints (Gleave et al., 2020; Cao et al., 2022); these experiments would make for interesting next steps.

ETHICS STATEMENT

We propose a novel framework to defend against adversarial attacks in RL setting by leveraging prior knowledge. This paper does not raise any ethical concerns. Our study does not involve human subjects. The Robot Foodcourt simulation environment we developed does not contain any sensitive information.

REPRODUCIBILITY STATEMENT

To ensure the reproducibility of our experimental results, we include detailed network architecture and hyper-parameters in the appendix and provide source code in the supplementary material.

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

## A TOY EXAMPLE: GOAL FINDING

To better illustrate how KPR works, we demonstrate it on a goal-finding grid-world environment as a proof of concept evaluation.

**Environment.** The agent's objective is to find a target goal while avoiding the obstacle in a $7 \times 7$ grid-world (Figure 4A.). The agent receives a reward of $+10$ for reaching the target, a $-10$ penalty for colliding with the obstacle, and a $-1$ penalty at each time step. The maximum episode length is 30. To mimic the redundant information in real-world scenarios, each object has a specific color, shape, and letter.

**Auxiliary Tasks and Domain Knowledge.** We design auxiliary tasks that focus on the different aspects of the objects, such as "$T_1$: Find the orange color (target object color)", "$T_2$: Avoid the diamond shape (obstacle object shape)". We denote the main task as $T_M$. A simple domain knowledge relation among them could be $T_M \implies T_1 \wedge T_2$. A complete list of auxiliary tasks and domain knowledge relations among them can be found in Appendix B.1.

**Compared Methods.** We compared with policy ensemble (Wiering & van Hasselt, 2008) and MLP Fusion which is a variant of KPR. Instead of leveraging GNN based pooling function to incorporate logical relations among auxiliary tasks, MLP Fusion uses MLP to replace the GNN pooling function. We aim to investigate the role of the logical relation domain knowledge component by comparing it with this variant. We use simple Deep Q-Learning (DQN) (Mnih et al., 2015) to train our main task and auxiliary tasks policies. Alternative algorithms and policies can be used without changing the overall proposed framework, e.g., improved DQN variants (Hasselt et al., 2016; Hessel et al., 2018) and alternative algorithms (Schaul et al., 2015; Bellemare et al., 2017; Christodoulou, 2019).

**Result and Discussion.** We adopt FGSM attack (Goodfellow et al., 2015) to perturb the observation. Since the observations are grid world states, the attacks here are generally stronger than the ones on pixel perturbations. The performance is evaluated in terms of the episode return averaged over 1000 episodes. The results are summarized in Figure 4B. C.. While the performance is similar in the Benign Environment, KPR outperforms others by a large margin in the perturbed environment. Note that KPR is not trained in the perturbed environment. This demonstrated KPR's ability to correct the inconsistency caused by adversarial attacks by leveraging domain knowledge over auxiliary policies.

Furthermore, to investigate the qualitative correlation between the amount of domain knowledge and episode return, we study how different knowledge levels affect the test performance, as shown in Figure 4 D. E.. The "Low" knowledge level consists of 4 auxiliary policies and 2 formulae. The "Moderate" knowledge level consists of 11 auxiliary tasks and 5 formulae. And the "High" knowledge level consists of 17 auxiliary tasks and 12 formulae. As shown in Figure 4 D. and E., as the knowledge level increases, the improved robustness to adversarial attacks also increases.

### A.1 AUXILIARY TASKS AND DOMAIN KNOWLEDGE

The objective for the agent is to find the goal while avoiding the obstacle in a $7 \times 7$ grid world (Figure 4A.). The agent gets a $+10$ reward for reaching the target, a $-10$ penalty for touching the obstacle, and a $-1$ penalty every time step. The maximum step number is 30. To mimic the redundant information in real-world scenarios, we use three features to represent a single object, color, shape, and the letter on it. The auxiliary tasks are:

1. Find the orange star "T" object.
2. Find the orange object.
3. Find the star object.
4. Find the "T" object.
5. Avoid the blue diamond "O" object.
6. Avoid the blue object.
7. Avoid the diamond object.
8. Avoid the "O" object.
9. Avoid the orange star "T" object (negation task).

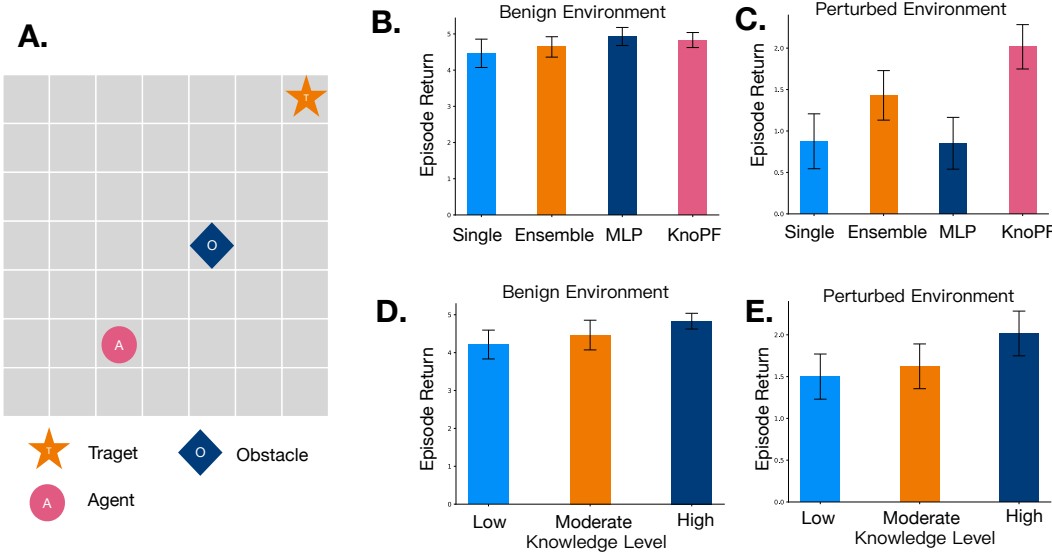

Figure 4: (**A.**) Goal finding grid world. (**B.** and **C.**) Test-time performance comparison. Episode return of main task policy, policy ensemble, MLP Fusion, and KPR in the goal finding grid-world environment. (**D.** and **E.**) The comparison of how the different knowledge levels, measured in terms of the number of auxiliary policies and formulae, affect the episode return. The episode returns are averaged over 1000 test episodes. Error bars indicate standard error.

10. Find the blue diamond "O" object (negation task).
11. Find the orange object and avoid the blue diamond "O" object.
12. Find the star object and avoid the blue diamond "O" object.
13. Find the "T" object and avoid the blue diamond "O" object.
14. Find the orange star "T" object and avoid the blue object.
15. Find the orange star "T" object and avoid the diamond object.
16. Find the orange star "T" object and avoid the "O" object.
17. Avoid the orange star "T" object and find the blue diamond "O" object.

There are many possible relations among them. The logical relations we adopted are:

1. $T_M \implies T_1 \wedge T_5$
2. $T_M \implies T_1 \wedge T_6$
3. $T_M \implies T_1 \wedge T_7$
4. $T_M \implies T_1 \wedge T_8$
5. $T_M \implies T_2 \wedge T_5$
6. $T_M \implies T_3 \wedge T_5$
7. $T_M \implies T_4 \wedge T_5$
8. $T_M \implies \neg T_{17}$
9. $T_M \implies (\neg T_9) \wedge (\neg T_{10})$
10. $T_M \implies (T_2 \vee T_3 \vee T_4) \wedge (T_6 \vee T_7 \vee T_8)$
11. $T_M \implies T_{11} \vee T_{12} \vee T_{13}$
12. $T_M \implies T_{14} \vee T_{15} \vee T_{16}$

where $T_M$ denotes the main task.

# B ATARI GAMES ENVIRONMENT

## B.1 AUXILIARY TASKS AND DOMAIN KNOWLEDGE

### B.1.1 ROAD RUNNER

The Road Runner runs endlessly to the left. The player must pick up bird seeds on the street, avoid cars (Wikipedia, 2022b). The auxiliary tasks are:

1. Collect bird seeds.

2. Avoid cars.

There are many possible relations among them. The logical relations we adopted are:

1. $T_M \implies T_1 \wedge T_2$

where $T_M$ denotes the main task.

### B.1.2 RIVER RAID

River Raid is a vertically scrolling shooter game. The player flies a fighter get which can only move left and right. It cannot maneuver up and down the screen, but it can accelerate and decelerate. The player's jet crashes if it collides with the riverbank or an enemy craft, or if the jet runs out of fuel. The player scores points for shooting enemy tankers (30 points), helicopters (60 points), fuel depots (80 points), jets (100 points), and bridges (500 points). The jet refuels when it flies over a fuel depot. A bridge marks the end of a game level (Wikipedia, 2022a). The auxiliary tasks are:

1. Shoot enemy tankers.

2. Shoot enemy helicopters.

3. Get fuel depots.

4. Shoot enemy jets.

5. Shoot enemy bridges.

6. Avoid collides with riverbank or enemy craft.

There are many possible relations among them. We sampled a subset:

1. $T_M \implies T_1 \wedge T_2 \wedge T_3 \wedge T_4 \wedge T_5 \wedge T_6$
2. $T_M \implies T_1 \wedge T_6$
3. $T_M \implies T_2 \wedge T_6$
4. $T_M \implies T_5 \wedge T_6$
5. $T_M \implies T_1 \wedge T_2$
6. $T_M \implies T_1 \wedge T_5$
7. $T_M \implies T_1 \wedge T_4$
8. $T_M \implies T_2 \wedge T_3$
9. $T_M \implies T_2 \wedge T_4$
10. $T_M \implies T_3 \wedge T_5$
11. $T_M \implies T_3 \wedge T_6$
12. $T_M \implies T_4 \wedge T_5$

where $T_M$ denotes the main task.

### B.1.3 SPACE INVADERS

Space Invaders is a fixed shooter in which the player moves a laser cannon horizontally across the bottom of the screen and fires at aliens overhead. The aliens begin as five rows of eleven that move left and right as a group, shifting downward each time they reach a screen edge. The goal is to eliminate all of the aliens by shooting them. While the player has three lives, the game ends immediately if the invaders reach the bottom of the screen. The aliens attempt to destroy the player's cannon by firing projectiles. The laser cannon is partially protected by stationary defense bunkers which are gradually destroyed from the top by the aliens and, if the player fires when beneath one, the bottom (Wikipedia, 2022c). The auxiliary tasks are:

1. Shoot the first row of aliens.

2. Shoot the second row of aliens.

3. Shoot the third row of aliens.

4. Shoot the forth row of aliens.

5. Shoot the fifth row of aliens.

6. Avoid being shot by aliens.

There are many possible relations among them. We sampled a subset:

1. $T_M \implies T_1 \wedge T_2 \wedge T_3 \wedge T_4 \wedge T_5 \wedge T_6$

2. $T_M \implies T_1 \wedge T_6$

3. $T_M \implies T_2 \wedge T_6$

4. $T_M \implies T_5 \wedge T_6$

5. $T_M \implies T_1 \wedge T_2$

6. $T_M \implies T_1 \wedge T_5$

7. $T_M \implies T_1 \wedge T_4$

8. $T_M \implies T_2 \wedge T_3$

9. $T_M \implies T_2 \wedge T_4$

10. $T_M \implies T_3 \wedge T_5$

11. $T_M \implies T_3 \wedge T_6$

12. $T_M \implies T_4 \wedge T_5$

where $T_M$ denotes the main task.

## C ROBOT FOOD COURT (ROFOCO) ENVIRONMENT

In order to evaluate the performance of KPR in a complex real-world like environment, we developed a high fidelity simulated Robot Food Court environment (RoFoCo) via Unity (Juliani et al., 2018), as shown in Figure 2. The agent is a food serving robot. A food stall number and a table number are instructed at the beginning of every trajectory. The main task is to collect food from the correct food stall and deliver it to the correct table number, then after the customer finish eating, pick up the used tray and send it to the tray collection point. The agent gets a $+10$ reward for completing each intermediate stage, and an additional $+20$ reward for completing the whole task. The agent will get a $-0.1$ penalty every time step and an additional $-5$ penalty if it tries to perform pick up or put down actions on objects that are not affordable, such as trying to pick up something on humans. The maximum number of steps is 1000. The available actions are: 1) Move forward; 2) Move backwards; 3) Turn left 90 degrees; 4) Turn Right 90 degrees; 5) Pick up; 6) Put down; 7) Do nothing. The observation is $128 \times 128$ RGB image.

### C.1 AUXILIARY TASKS AND DOMAIN KNOWLEDGE

The auxiliary tasks are:

1. Navigate to food stall number $x$.
2. Navigate to customer table number $x$.
3. Navigate to tray collection point.
4. Pick up food from food stall number $x$.
5. Put down food on customer table number $x$.
6. Pick up used tray from customer table number $x$.
7. Put down used tray to tray collection point.
8. Pick up food from customer table number $x$ (negation task).
9. Put down used tray on customer table number $x$ (negation task).
10. Put down food on food stall number $x$ (negation task).
11. Put down food on tray collection point (negation task).
12. Put down used tray on food stall number $x$ (negation task).

There are many possible relations among them. The logical relations we adopted are:

1. $T_M \implies T_1 \wedge T_4$
2. $T_M \implies T_2 \wedge T_5$
3. $T_M \implies T_2 \wedge T_6$
4. $T_M \implies T_3 \wedge T_7$
5. $T_M \implies T_1 \wedge T_4 \wedge (\neg T_{10}) \wedge (\neg T_{12})$
6. $T_M \implies T_2 \wedge T_5 \wedge (\neg T_9)$
7. $T_M \implies T_2 \wedge T_6 \wedge (\neg T_8)$
8. $T_M \implies T_3 \wedge T_7 \wedge (\neg T_{11})$
9. $T_M \implies T_1 \wedge T_4 \wedge (\neg T_8)$
10. $T_M \implies T_2 \wedge T_5 \wedge (\neg T_{10})$
11. $T_M \implies T_3 \wedge T_7 \wedge (\neg T_9)$
12. $T_M \implies \neg T_8$
13. $T_M \implies \neg T_9$
14. $T_M \implies \neg T_{10}$
15. $T_M \implies \neg T_{11}$
16. $T_M \implies \neg T_{12}$

where $T_M$ denotes the main task.

## D ADVERSARIAL ATTACKS

We evaluated against 3 white-box attacks, FGSM, PGD, Jitter, and 1 black-box attack, Square attack. We'll briefly introduce their intuitions and mechanisms, as well as the hyper-parameters we use. To ensure fair comparison, we utilize the same $\epsilon$ value, which controls the maximum perturbation value, for all 4 adversarial attacks. According to different task characteristics, we use 0.002 for Road Runner, 0.01 for River Raid, 0.002 for Space Invaders and 0.002 for Robot Foodcourt environment. Pixel values are scaled to $[0, 1]$.

## D.1 FAST GRADIENT SIGN METHOD (FGSM)

FGSM (Goodfellow et al., 2015) is a method to efficiently calculate the gradient of the cost function with respect to the input of the neural network. The adversarial examples are generated using the following equation:

$$x' = x + \epsilon \cdot \text{sign}(\nabla_x J(\theta, x, y)) \tag{9}$$

where $\theta$ denotes the parameters of a model, $x$ is the input to the model, $y$ is the target associated with $x$, $J(\theta, x, y)$ is the cost used to train the neural network, and `sign` is the component-wise signum operator. The adversarial examples generated by FGSM exploit the "linearity" of deep network models in the higher dimensional spaces whereas such models were commonly thought to be highly non-linear at that time. They hypothesized that the designs of deep neural networks that encourage linear behavior for computational gains also make them susceptible to cheap analytical perturbations, which is often referred as "linearity hypothesis".

## D.2 PROJECTED GRADIENT DESCENT (PGD)

PGD (Madry et al., 2018) improve the performance of FGSM by running a finer iterative optimizer for multiple iterations. PGD performs FGSM with a smaller step size and projects the updated adversarial sample into the $\epsilon - L_\infty$ neighbor of the benign samples and a valid range. Hence the adversarial perturbation size is smaller than $\epsilon$. The update procedure follows:

$$x'_{t+1} = \text{Proj}\{x'_t + \alpha \cdot \text{sign}[\nabla_x J(\theta, x'_t, y)]\} \tag{10}$$

## D.3 JITTER ATTACK

In order to make adversarial attacks more effective, Jitter (Schwinn et al., 2021) proposes a novel loss function to encourage logits scale invariance, diverse attack targets, and perturbation norm minimization. The final loss function can be described as follows:

$$\mathcal{L}_{\text{Jitter}} = \begin{cases} \frac{\|\hat{z} - y + \mathcal{N}(0, \sigma)\|_2}{\|\gamma\|_p} & \text{if } x' \text{ is misclassified} \\ \|\hat{z} - y + \mathcal{N}(0, \sigma)\|_2 & \text{if } x' \text{ is not misclassified yet} \end{cases}$$

$$\hat{z} = \text{softmax}(\alpha \cdot \frac{z}{\|z\|_\infty}) \tag{11}$$

where $y$ is the ground truth, $z$ is the output logits after perturbation, and $\alpha$ controls the lowest and largest possible output values of the softmax function.

## D.4 SQUARE ATTACK

Square attack (Andriushchenko et al., 2020) is based on a randomized search scheme which selects localized square-shaped updates at random positions so that at each iteration the perturbation is situated approximately at the boundary of the feasible set. The objective is to solve the constrained optimization problem:

$$\min_{\hat{x} \in \mathcal{S}} \mathcal{L}(f(\hat{x}), y) = f_y(\hat{x}) - \max_{k \neq y} f_k(\hat{x}), \quad \text{s.t.} \quad \|\hat{x} - x\|_p \leq \epsilon \tag{12}$$

where $f$ is the target network, $x$ is the input, $y$ is the ground truth, and $\mathcal{S}$ is the domain of the input.

## E MODEL ARCHITECTURE

**Policy Network.** We adopt PPO (Schulman et al., 2017; Huang et al., 2022) as the learning algorithm. We use a three-layer CNN with $\{32, 64, 64\}$ hidden size, followed by a two linear layers with $512$ neurons for both actor and critic networks.

**VAE.** The encoder is five-layer CNN with hidden dimension $\{32, 64, 128, 256, 512\}$, followed by a linear layer. The decoder is symmetric to the encoder with an additional 2D transposed convolution layer (Zeiler et al., 2010) of hidden dimension $512$.

**GNN Fusion Network.** Our GNN pooling network consists of 3 GCN convolution layers with hidden dimensions $\{64, 32, 64\}$, followed by the graph mutliset transformer as described in Section 3, and 2 linear layers of $64$ hidden neurons.

**MLP Fusion Network** The MLP Fusion Network consists of 2 linear layers of $64$ hidden neurons.

