# OpenReview forum: "Don't Throw Your Old Policies Away: Knowledge-based Policy Recycling Protects Against Adversarial Attacks"
_ICLR.cc/2023/Conference — Submitted to ICLR 2023_

### Official Review · Reviewer_XTBx · 2022-10-24

**Confidence:** 3
**Correctness:** 4
**Technical Novelty And Significance:** 2
**Empirical Novelty And Significance:** 3
**Recommendation:** 6

**Clarity, Quality, Novelty And Reproducibility:**

Quality: This work aims to solve the robustness of RL models and provides a flexible framework to encode domain knowledge into the RL training procedure. And experiments are extensive and strongly support the author’s claims. In short, the quality of this paper is high.

Clarity: The paper is well written and pretty clear in demonstrating the ideas and how the framework details are implemented. The idea is easy to follow, but since the code is not provided, the evaluation is neutral.

Originality: This paper is extended based on past work in prediction tasks which weakens its novelty. But the extension is nontrivial, and the proposed framework is flexible and easy to apply to different RL agents. So the novelty is neutral.


**Strength And Weaknesses:**

Strength:

- The author is trying to introduce methods used in prediction tasks into reinforcement learning and solving the problem of the adversarial attack, which is a very important limitation hindering RL implementation in the real world. The idea is pretty concise and focused on an interesting task.

- Using the past policies or sub-tasks policies and domain knowledge to help improve the RL model's robustness can help reduce the implementation cost and explainability of RL models. Also, the proposed framework is very flexible and easily extended to different agents.
The experiments support the author's claims and are extensively investigated in different scenarios. And the proposed methods uniformly outperform other baseline models.

Weakness:


- Part of this paper’s idea is extended from past work on prediction tasks. The basic idea is using assembly learning and combining the idea from past prediction tasks as constraints on the policies, which slightly weakens the novelty of this paper.

- The baseline model seems not to involve adversarial detection approaches, and in Table 1, the performance of MLP Fusion and KPR even increased after applying Square adversarial attacks. It is strange but not discussed by the author.


**Summary Of The Paper:**

In this paper, the author extended using domain knowledge to improve adversarial attacks from supervised learning to reinforcement learning and makes learned main policy universally resistant to adversarial attacks without prior knowledge of the attack.

The main contributions consist of 1. It generalizes previous work, which encodes domain knowledge as logical formula and a set of features and adds to the objective to apply constraints on the learned policies close to logical formulae from prediction tasks to reinforcement learning approaches. 2. The new extended approach is first used to tackle adversarial attack problems in reinforcement learning (RL), which is very important to keep RL performing appropriately under noisy input signals. 3. It is the first paper investigating using domain knowledge to defend against adversarial attacks in the form of policies.


**Summary Of The Review:**

This paper focuses on a very interesting domain which is the robustness of RL models under adversarial attacks, and the extended framework from the past prediction tasks is decent and easy to apply to various RL agents. Through the experiment results, this proposed method clearly outperforms the other baseline models universally. Thus I consider it a high-quality paper.

---

> ### Author Response · Authors · 2022-11-17
> **Response to Reviewer XTBx**
>
> We thank Reviewer WBRZ for the positive comments and for providing thoughtful feedback on our work. We provide additional details on specific comments below.
> > “Part of this paper’s idea is extended from past work on prediction tasks.”
>
> We are different from the supervised learning setting because the relations are over policies, not actions (classes). In the supervised learning setting, the domain knowledge is defined over classes and features (auxiliary classes). For example, if the class is "stop sign," with high probability, it is "octagon." The domain knowledge component will adjust the class prediction score if the detected feature is "not octagon."
>
> However, the extension to the reinforcement learning domain is non-trivial because the relations between the actions are not static. For example, the agent is supposed to perform differently, conditioned on different states. Furthermore, optimal actions for each state are still being determined, and rewards are generally sparse. Therefore, we propose to define domain knowledge over policies to enable high-level domain knowledge incorporation in sequential decision-making. More specifically, we fuse action distributions of the main task policy and auxiliary task policies according to their structural relation.
>
> > “the performance of MLP Fusion and KPR even increased after applying Square adversarial attacks. It is strange but not discussed by the author.”
>
> Although the means are slightly higher in some cases, the differences are not significant as the standard error ranges overlap.

---

### Official Review · Reviewer_EmJQ · 2022-10-24

**Confidence:** 4
**Correctness:** 2
**Technical Novelty And Significance:** 3
**Empirical Novelty And Significance:** 3
**Recommendation:** 3

**Clarity, Quality, Novelty And Reproducibility:**

The clarity of the writing is generally good, although it took a couple of times reading through Section 3 to fully understand all the pieces. One suggestion that might help improve clarity is to refer to the parts of Figure 1 throughout Section 3 as each piece of the method is introduced such that readers can follow along visually.

As far as originality, it seems like the method is fairly novel. I'm not so familiar with related adversarial defenses based on domain knowledge, so it's hard for me to judge how similar or different this paper's approach is.

I am most concerned about reproducibility. Besides not providing code, there are also a number of important pieces of information missing from the paper that would be necessary to reproduce the results:
 * The details of how the white-box attacks are applied to the RL domain (see first bullet above in strengths and weaknesses)
 * Parameters of attacks like step size, number of iterations, norm bound, etc.
 * Details of neural network architectures
 * Hyperparameters for RL

**Strength And Weaknesses:**

Overall, much of the paper is well-written and the experiments show promising results. However, I have a couple major concerns about the evaluation:

 * First, the evaluation seems to be underspecified and thus not reproducible. The authors say they use standard white-box adversarial attacks like FGSM and PGD to attack policies. However, these algorithms cannot actually directly solve the attack optimization problem given in (2). This is because (2) is a function of the return of the policy, and one cannot directly differentiate the return with respect to the adversarial perturbations. In general, optimizing equation (2) would require more RL-like methods to approximate the gradient rather than the autograd methods used for typical whitebox attacks in supervised learning. Since the authors do not specify how they resolve this contradiction, I'm don't think the evaluation is reproducible. There are not further details in the appendix and there is no code, either.
 * Second, I'm also worried about gradient masking [1] in the evaluation. In particular, the aggregation mechanism given in equation (9) is not actually differentiable with respect to the underlying policies! This means that there are no gradients that an attacker can use to optimize the state towards lower reward. I'm not sure how the attacks are expected to work given this, and this makes me distrust the evaluation. While blackbox attacks should avoid this problem, looking at the results it appears that KPR does not show as large an improvement over the main task policy against black-box attacks, indicating that gradient masking may be occurring.
 * Third, it does not seem like the authors ran the entire training procedure using multiple different random seeds for the evaluation. This is a standard practice in RL evaluation and is needed to reduce the vast variance in RL outcomes across random seeds.

Given these three concerns, I think the evaluation needs to be improved before the proposed method can be reliably judged to perform better than the baselines. I encourage the authors to precisely report how the white-box attacks optimize equation (2), avoid gradient masking, and report results across multiple training runs with different random seeds.

Besides the evaluation concerns, there are some other downsides to the method, like the need for auxiliary tasks and logical relations between them.

[1] Athalye et al. Obfuscated Gradients Give a False Sense of Security: Circumventing Defenses to Adversarial Examples. ICML 2018.

**Summary Of The Paper:**

This paper focuses on the problem of adversarial attacks against deep RL policies. They propose a defense that uses an ensemble policy based on policies for various auxiliary tasks and logical relations between those auxiliary tasks and the main task. A graph neural network is used to combine action distributions from the various auxiliary policies over the graph induced by each logical relation. Then, policies are aggregated across all logical relations via a voting mechanism. In experiments in three Atari games and a simulated robotics environment, the aggregated policy seems to be more robust to various adversarial attacks.

**Summary Of The Review:**

Overall, I think this paper explores a promising direction but needs a fair amount of work to improve the evaluation and reproducibility. Therefore, I don't think it is ready for publication at ICLR. However, if the authors can provide a more convincing evaluation along with all the details and code to reproduce the experiments, I am happy to raise my score.

---

> ### Author Response · Authors · 2022-11-17
> **Response to Reviewer EmJQ**
>
> We thank Reviewer EmJQ for reviewing our paper and providing constructive feedback on our work.
> > “optimizing equation (2) would require more RL-like methods to approximate the gradient”
>
> Thank you for pointing this out. Equation (2) was meant as a general "catch-all" for adversaries that attempt to reduce the cumulative reward of the agent. That said, we agree that this formulation does not coincide well with attacks that are used in adversarial RL and in our experiments; we have followed prior work [2, 3, 4] that use similar supervised attacks since optimizing (2) explicitly is difficult due the large action space of pixel perturbations. To avoid confusion, we have dropped eq (2) and assume the adversary is attempting to change the predicted action of the agent.
>
> > “I'm also worried about gradient masking [1] in the evaluation. In particular, the aggregation mechanism given in equation (9) is not actually differentiable with respect to the underlying policies”
>
> Thanks for pointing out this relevant work. The gradient we get might not be globally “correct” due to the voting step. We will evaluate the attack proposed in [1]. Due to time constraints, we are still working on it and will get back to you once the results are ready.
>
> > “Besides the evaluation concerns, there are some other downsides to the method, like the need for auxiliary tasks and logical relations between them.”
>
> Our primary motivation is that auxiliary tasks and relations are available in many real-world scenarios. For example, 1) we want the home robot to make a latte. It may also know how to make espresso and how to steam milk. 2) we ask a cooking robot to make steaks. This involves controlling the doneness and seasoning at the right time; 3) we want the autonomous driving agent to drive in the rain, and it may know how to drive in sunny weather, fog, and snow. These are the intermediate or related task policies that are usually available in real-world scenarios. Therefore, we focus on how to use them instead of obtaining or learning them. Our work demonstrated that leveraging such auxiliary task policies that are available is able to enhance the robustness of the main task policy.
>
> One potential application of our method is in curriculum learning. In curriculum learning, intermediate policies are trained progressively. Instead of throwing them away, our work suggests potential usage for them to defend against policy attacks.
>
> > “I am most concerned about reproducibility.”
>
> We have included our source code in the supplementary material.
>
> [1] Athalye et al. Obfuscated Gradients Give a False Sense of Security: Circumventing Defenses to Adversarial Examples. ICML 2018.
>
> [2] Sandy Huang, Nicolas Papernot, Ian Goodfellow, Yan Duan, and Pieter Abbeel. Adversarial attacks on neural network policies. arXiv, 2017.
>
> [3] Vahid Behzadan and Arslan Munir. Vulnerability of Deep Reinforcement Learning to Policy Induction Attacks. In Machine Learning and Data Mining in Pattern Recognition, volume 10358, pp. 262–275. Springer International Publishing, Cham, 2017.
>
> [4] Anay Pattanaik, Zhenyi Tang, Shuijing Liu, Gautham Bommannan, and Girish Chowdhary. Robust deep reinforcement learning with adversarial attacks. In Proceedings of the 17th International Conference on Autonomous Agents and MultiAgent Systems, AAMAS ’18, 2018.

---

> > ### Comment · Reviewer_EmJQ · 2022-11-22
> > **Evaluation of fusion model in code is not correct**
> >
> > Thank you for the response. Here are a few points in reply to your specific comments:
> >  * While you've removed equation (2) which was confusing, you still don't specify in the paper what loss function the adversarial attacks are optimizing. This is an important detail since RL adversarial attacks are not standardized enough to assume a particular loss, and the RL setting is quite different from the supervised learning setting where the adversarial attacks you use were developed. Are you maximizing the cross-entropy of the policy to the original action it would have taken? Or maximizing the KL divergence between the original action distribution and the one under the perturbed observation? This should be clarified for reproducibility.
> >  * With respect to the voting aggregation mechanism, I've looked through your code and it looks like you are only attacking a single model of the ensemble in effect! If we look at the implementation of `Get_action` in `fusion_adv.py`, the logits returned from the forward pass are from the `out` variable in `cal`, which for the fusion model are set on line 433. This is overwritten as the surrounding for loop goes through each ensemble member, meaning that it only ends up holding the results of the last ensemble member. This probably severely reduces the strength of the attack for the fusion model and could lead to misleading results.
> >
> > Overall, since the attack is poorly implemented for the proposed method, I don't believe the results that suggest it is more robust than the baselines. One possible fix would be to attack all members of the ensemble (as opposed to just the last one), but even this does not directly optimize for changing the aggregated action because the aggregation is done in a non-differentiable way. I don't think there is enough time to implement this fix anyways before the end of the discussion period. Therefore, I cannot support this paper for acceptance since the evaluation is not correct. Please let me know if there's something in the code that I'm not understanding correctly.

---

### Official Review · Reviewer_dRJe · 2022-10-25

**Confidence:** 5
**Correctness:** 2
**Technical Novelty And Significance:** 2
**Empirical Novelty And Significance:** 1
**Recommendation:** 3

**Clarity, Quality, Novelty And Reproducibility:**

Clarity: sections 1-3 of the paper are well-written; manuscript clarity/quality somewhat declines afterwards. See above on issues with experimental design.

The novelty of the manuscript is limited to its problem formulation, but even this relies on strong assumptions; I found neither the problem definition nor the proposed methodology particularly compelling.

No major issues with reproducibility, although I would recommend capturing the main hyperparameters in the main content of the manuscript.

**Strength And Weaknesses:**

(Strengths)

I like the notion of combining domain knowledge with learning-based methodology.

Sections 1-3 are pretty well-written.

(Weaknesses)

Section 3.1: What are these success thresholds and how are they obtained, for an arbitrary collection of tasks and attacks? This seems non-trivial to define and combinatorial in complexity.

Section 3.2: Architecturally, the proposed approach is very straightforward, and I was hard-pressed to find much novelty there.

Section 4: It would be helpful if the manuscript provided the full names, brief descriptions, and some intuition behind why specific attacks were chosen.

Section 4.1: In many practical applications, the subtasks/objectives can be conflicting — how are these situations reconciled?

Section 4.1, Section 4.2: I am not really convinced by the experiments that the manuscript chose to perform. I would have liked to see empirical results on more standardised or challenging evaluation benchmarks in RL — especially those that include aspects of safety or robustness, such as OpenAI Safety Gym, Learn-to-Race, Carla Challenge, etc. Discussion of adversarial attacks in these tasks, where subtask objectives are often conflicting (!), would be much more compelling and intuitive.

Section 4.1: It would be helpful to show the cumulative episodic reward curves (with error areas), by repeatedly evaluating on the test environment after a fixed number of training iterations or environment interactions: this would also give intuition on the training dynamics and sample-complexity.

**Summary Of The Paper:**

The manuscript proposes an approach for combining domain knowledge with RL; the methodology encodes propositional logical rules via Graph Neural Networks, with multiple pooling and attention mechanisms.

**Summary Of The Review:**

The clarity of the manuscript declines in its experimental design and adversarial attacks illustration.

The problem formulation and methodology rely on strong assumptions.

Missing discussion of limitations, ethics, and reproducibility statements.

---

> ### Author Response · Authors · 2022-11-17
> **Response to Reviewer dRJe**
>
> We thank Reviewer dRJe for their detailed comments. We commit to addressing minor suggestions. In the following, we discuss the main issues raised:
> > “Section 3.1: What are these success thresholds and how are they obtained, for an arbitrary collection of tasks and attacks? This seems non-trivial to define and combinatorial in complexity.”
>
> We do not need to define any thresholds. For example, A and B -> M means if both tasks A and B are successful, then the main task M is also successful. We will optimize the reward of the main task and do not need to define specific thresholds for any auxiliary tasks.
>
> > “Section 3.2: Architecturally, the proposed approach is very straightforward, and I was hard-pressed to find much novelty there.”
>
> Our novelty lies in knowledge integration instead of architectural design. The architecture was intentionally chosen to be simple for implementation and usage.
>
> > “Section 4: It would be helpful if the manuscript provided the full names, brief descriptions, and some intuition behind why specific attacks were chosen.”
>
> Thanks for the suggestions. We have included it in appendix D.
>
> > “Section 4.1: In many practical applications, the subtasks/objectives can be conflicting — how are these situations reconciled?”
>
> There are conflicting auxiliary tasks in our work. We add the “negation” logical operator in front of the negative task. Our fusion GNN model is trained to learn different logical operators. However, we assume there is only one main task that we would like to optimize.
>
> > “Missing discussion of limitations, ethics, and reproducibility statements.”
>
> We have added them to the paper.
>
> > “The novelty of the manuscript is limited to its problem formulation, but even this relies on strong assumptions; I found neither the problem definition nor the proposed methodology particularly compelling.”
>
> Our primary assumption is that it is harder for the attacker to perturb such that all the auxiliary tasks are altered consistently according to our domain knowledge. We think the assumptions hold in general scenarios. Which assumptions do you think are too strong? We could discuss them further.
>
> > “I am not really convinced by the experiments that the manuscript chose to perform”
>
> We evaluated our model in Atari game environments and a food-serving robot foodcourt environment. We choose Atari games because it is a standard benchmark used in prior work on adversarial attacks in reinforcement learning[1, 2, 3, 4]. The robot foodcourt environment was chosen as a complex realistic domain which has a decomposable step-by-step task, which naturally involves auxiliary tasks and common sense relations between the auxiliary tasks.
>
> [1] Sandy Huang, Nicolas Papernot, Ian Goodfellow, Yan Duan, and Pieter Abbeel. Adversarial attacks on neural network policies. arXiv, 2017.
>
> [2] Vahid Behzadan and Arslan Munir. Vulnerability of Deep Reinforcement Learning to Policy Induction Attacks. In Machine Learning and Data Mining in Pattern Recognition, volume 10358, pp. 262–275. Springer International Publishing, Cham, 2017.
>
> [3] Huan Zhang, Hongge Chen, Chaowei Xiao, Bo Li, Mingyan Liu, Duane Boning, and Cho-Jui Hsieh. Robust deep reinforcement learning against adversarial perturbations on state observations. In Advances in Neural Information Processing Systems, volume 33, pp. 21024–21037, 2020.
>
> [4] Yanchao Sun, Ruijie Zheng, Yongyuan Liang, and Furong Huang. Who is the strongest enemy? towards optimal and efficient evasion attacks in deep RL. In International Conference on Learning Representations, 2022.

---

### Official Review · Reviewer_Vu3r · 2022-10-25

**Confidence:** 4
**Correctness:** 3
**Technical Novelty And Significance:** 3
**Empirical Novelty And Significance:** 2
**Recommendation:** 5

**Clarity, Quality, Novelty And Reproducibility:**

Clarity: The author has a very clear display of the structure of KPR, and the description of the experimental settings and details is also relatively clear.
Quality: The charts are well made, and the visual effects of the experimental environment constructed are also very good.
Novelty: KPR is different from previous methods, this is the first work to demonstrate that domain knowledge in the form of policies can be used to defend against adversarial attacks.
Reproducibility: The author did not provide the source code, so I cannot confirm it


**Strength And Weaknesses:**

a)	Strength:
The authors describe in detail the differences between KPR and previous methods:
policy ensemble, policy distillation, and adversarial training, and designed a very sufficient experiment to verify the advantages of KPR. To investigate the role domain knowledge plays, authors included a variant of KPR, MLP Fusion, which replaces the GNN with an MLP whose input is the action distributions of the auxiliary policies. This ablation study is convincing and shows the effect of domain knowledge. In the experimental part, the choice of experimental environment is very interesting, the visual effect is great, and the experimental data is relatively sufficient, showing the advantages of KPR in the form of a table
b)	Weaknesses:
The author makes a strong assumption that the susceptibility of a policy to an attacker is due principally to overfitting on a specific task. If only one specific type of attack is considered, it is not enough to demonstrate the generalization of defense strategies and the use of GNN for a well-structured use of domain knowledge. In terms of theory, there seems to be a lack of more analysis. The author needs to explain why this is necessary. The theoretical analysis in methods such as fuzzy control can have a fuzzy mathematical system to describe how to convert from natural language to numerical output.


**Summary Of The Paper:**

This paper proposed a new framework called Knowledge-based Policy Fusion (KPR), which leverages domain knowledge to defend against adversarial attacks in RL. KPR incorporates domain knowledge from auxiliary policies and specified logical relations between tasks, then learns flexible relations from interaction data via graph neural networks. Different from prior defense methods in reinforcement learning, such as adversarial training and robust learning, the main advantage of KPR is that it is both policy and attack agnostic; any type of policy could be utilized, and no access nor information about the attack is required. They demonstrated its efficacy empirically in both Atari games and the complex Robot Food Court environment (RoFoCo). According to the authors, this is the first work to demonstrate that domain knowledge in the form of policies can be used to defend against adversarial attacks.

**Summary Of The Review:**

This paper proposed a new framework called KPR, which leverages domain knowledge to defend against adversarial attacks in RL. The charts are clear and easy to understand, the descriptions are also very concise and clear, and the experiments are sufficient, but the scenario assumptions are single, and some theoretical explanations are lacking

---

> ### Author Response · Authors · 2022-11-17
> **Response to Reviewer Vu3r**
>
> We thank Reviewer Vu3r for reviewing our work and hope that the responses adequately address their concerns.
> > “one specific type of attack is considered, it is not enough to demonstrate the generalization of defense strategies and the use of GNN for a well-structured use of domain knowledge.”
>
> We are unable to understand this concern and would appreciate additional clarification/elaboration. We do not consider one specific type of attack and evaluated our proposed method using a set of common white-box and black-box attacks.
>
> > “The theoretical analysis in methods such as fuzzy control can have a fuzzy mathematical system to describe how to convert from natural language to numerical output.”
>
> Thanks for the constructive suggestion. Although theoretical work in this domain is important, it is not the focus of our work. Given our setup, it is unclear how theoretical results can be obtained without unrealistically strong assumptions. We have examined fuzzy control but it is unclear how this setting can be used to theoretically analyze our system.

---

### Author Response · Authors · 2022-11-17
**The collective response to all reviewers**

We thank the reviewers for their thoughtful and constructive review of our work. We are encouraged to hear that the reviewers found our idea of leveraging domain knowledge to defend against policy attacks interesting and novel (Reviewers Vu3r, EmJQ, XTBx) and that the presentation was clear (Reviewer dRJe, EmJQ).

In response to feedback, we answer specific points in individual responses to each reviewer below. In addition, we have made the following changes:

- Added adversarial attacks description in Appendix D.
- Added model architecture description in Appendix E.
- Added ethics and reproducibility section.
- Included source code in the supplementary material.

---

### Decision · Program_Chairs · 2023-01-20

**Decision:**

Reject

**Justification For Why Not Higher Score:**

The reviewers pointed out several weaknesses in the paper, and there was a consensus for rejection.

**Justification For Why Not Lower Score:**

N/A

**Metareview: Summary, Strengths And Weaknesses:**

The reviewers agreed that the paper proposes a novel framework for leveraging domain knowledge to defend against adversarial attacks, is generally well-written, and results are promising. However, the reviewers pointed out several weaknesses in the paper and shared common concerns. We want to thank the authors for their detailed responses. Based on the reviewers' concerns and follow-up discussions, unfortunately, the final decision is a rejection. The reviewers have provided detailed and constructive feedback. We hope the authors can incorporate this feedback when preparing future revisions of the paper.